# Cerebral Insulin Bolus Revokes the Changes in Hepatic Lipid Metabolism Induced by Chronic Central Leptin Infusion

**DOI:** 10.3390/cells10030581

**Published:** 2021-03-06

**Authors:** Vicente Barrios, Elena López-Villar, Laura M. Frago, Sandra Canelles, Francisca Díaz-González, Emma Burgos-Ramos, Gema Frühbeck, Julie A. Chowen, Jesús Argente

**Affiliations:** 1Department of Endocrinology, Hospital Infantil Universitario Niño Jesús, Instituto de Investigación La Princesa, E-28009 Madrid, Spain; elena.lopez.villar@gmail.com (E.L.-V.); laura.frago@uam.es (L.M.F.); sandra.canelles@salud.madrid.org (S.C.); julieann.chowen@salud.madrid.org (J.A.C.); 2Centro de Investigación Biomédica en Red de Fisiopatología de la Obesidad y Nutrición (CIBEROBN), Instituto de Salud Carlos III, E-28029 Madrid, Spain; gfruhbeck@unav.es; 3Department of Pediatrics, Faculty of Medicine, Universidad Autónoma de Madrid, E-28029 Madrid, Spain; 4Institute of Medical and Molecular Genetics (INGEMM), IdiPAZ, Hospital Universitario La Paz, Universidad Autónoma de Madrid, E-28049 Madrid, Spain; francisca.diaz.gonzalez@idipaz.es; 5Faculty of Environmental Sciences and Biochemistry, Universidad de Castilla-La Mancha, E-45071 Toledo, Spain; emma.burgos@uclm.es; 6Metabolic Research Laboratory, Clínica Universidad de Navarra, E-31008 Pamplona, Spain; 7IMDEA Food Institute, CEI UAM + CSIC, E-28049 Madrid, Spain

**Keywords:** growth hormone axis, insulin signaling, leptin, lipid metabolism, liver

## Abstract

Central actions of leptin and insulin on hepatic lipid metabolism can be opposing and the mechanism underlying this phenomenon remains unclear. Both hormones can modulate the central somatostatinergic system that has an inhibitory effect on growth hormone (GH) expression, which plays an important role in hepatic metabolism. Using a model of chronic central leptin infusion, we evaluated whether an increase in central leptin bioavailability modifies the serum lipid pattern through changes in hepatic lipid metabolism in male rats in response to an increase in central insulin and the possible involvement of the GH axis in these effects. We found a rise in serum GH in leptin plus insulin-treated rats, due to an increase in pituitary GH mRNA levels associated with lower hypothalamic somatostatin and pituitary somatostatin receptor-2 mRNA levels. An augment in hepatic lipolysis and a reduction in serum levels of non-esterified fatty acids (NEFA) and triglycerides were found in leptin-treated rats. These rats experienced a rise in lipogenic-related factors and normalization of serum levels of NEFA and triglycerides after insulin treatment. These results suggest that an increase in insulin in leptin-treated rats can act on the hepatic lipid metabolism through activation of the GH axis.

## 1. Introduction

Leptin is a central mediator of endocrine circuits that regulate energy homeostasis providing feedback information to the hypothalamus regarding the energy status and decreasing body weight [1]. Insulin is also a critical regulator of energy balance, acting in the same hypothalamic areas as leptin to suppress feeding [2]. Leptin also regulates energy homeostasis by acting in peripheral tissues, with central administration modulating peripheral metabolism through changes in insulin sensitivity [3].

The liver is a key organ in lipid metabolism, with leptin reducing hepatic lipid content by decreasing the activity of lipogenic enzymes and up-regulating the expression of those implicated in lipolysis [4]. Insulin promotes lipogenesis through activation of fatty acid synthase (FAS) and other enzymes of lipid anabolism [5]. Additionally, insulin increases levels of hepatic malic enzyme [6] as well as the activity of glucose-6-phosphate dehydrogenase (G6PD) in the liver [7]. Importantly, an increase in the levels and activity of these “lipogenic” enzymes is crucial to supply the reduced form of nicotinamide adenine dinucleotide phosphate (NADPH) for lipogenesis [8].

High brain insulin sensitivity is associated with weight loss and related to a favorable fat distribution [9], whereas insulin resistance is linked to visceral adiposity and pathological accumulation of intrahepatic fat content [10]. Our findings indicate that an increase in central insulin bioavailability may modify synthesis and accumulation of fat in the liver, as well as provoke changes in adipose tissue that could affect systemic lipid concentrations.

Central leptin exerts a stimulatory effect on growth hormone (GH) secretion [11,12] suggesting that an increase in central insulin bioavailability may be essential for activation of the GH axis, given that insulin potentiates hypothalamic leptin functions [13]. The somatostatin (SRIF) system inhibits pituitary function, mostly reducing hormone secretion [14], although it also reduces GH mRNA content [15,16]. Leptin inhibits SRIF expression and secretion [17], and in turn, brain delivery of SRIF or SRIF receptor agonists diminish hypothalamic leptin signaling [18].

GH deficiency and insulin resistance are risk factors for the development of nonalcoholic steatohepatitis and previous reports suggest a relationship between these hormones in this pathology [19,20]. GH regulates numerous metabolic processes in the liver, including lipid metabolism via interactions with leptin and insulin [21,22] and promotion of triglyceride uptake [23]. In fact, maintenance of normal insulinemia seems to be required to maintain physiological hepatic GH sensitivity [24]. GH is also implicated in the generation of NADPH for lipogenesis, as up-regulation of the GH axis correlates with augmented hepatic G6PD expression [25]. Short-term effects of GH are insulin-like, inhibiting lipolysis and stimulating lipogenesis, whereas prolonged elevated concentrations of this hormone provoke insulin resistance related to reduction in phosphatidylinositol-3 kinase (PI3K) activation [26].

These previous data suggest that central leptin and insulin infusion can be involved in changes in the GH axis, with subsequent effects of this hormone on hepatic lipid metabolism. The current study was designed to determine the effects of an increase in central leptin and insulin bioavailability on hepatic lipid metabolism and its influence on the se-rum lipid profile. We examined the possible influence of changes in GH levels on these parameters, as well as some of the central mechanisms that could modulate GH expression after insulin infusion. To differentiate between the effects of leptin and those due only to a reduction in food intake, a group of pair-fed rats was included. We also studied the effect of central leptin and insulin infusion in adipose tissue, to analyze its possible contribution to the serum lipid profile. Finally, as central leptin infusion causes an increase in serum leptin levels [27], the effects of chronic peripheral leptin administration on these parameters were also analyzed.

## 2. Materials and Methods

### 2.1. Animals

All procedures were carried out in accordance with the local ethics committee and complied with Royal Decree 53/2013 pertaining to the protection of experimental animals and with the European Communities Council Directive (2010/63/EU). This study was approved by the Ethical Committee of Animal Experimentation of the Universidad de Alcalá (PROEX018/16, 14 June 2016). All care was taken to use the minimum number of animals. Male Wistar rats (250 ± 10 g) purchased from Harlan Laboratories (Barcelona, Spain) were individually caged and fed standard chow and water ad libitum. Animals were anesthetized using 4 mg of ketamine/100 g body weight (bw) and 0.5 mg of xylazine/100 g bw throughout chirurgical procedures.

### 2.2. Experimental Design

#### 2.2.1. Central Leptin and Insulin Infusion

Thirty adult male Wistar rats (250 ± 10 g) were treated with central leptin and/or insulin as previously published [27]. Briefly, a cannula attached to an osmotic minipump (Alzet, Durect Corporation, Cupertino, CA, USA) containing either saline or leptin (Preprotech, Rocky Hill, NJ, USA; delivering 12 μg/day at a volume of 0.5 µl/h) was implanted and maintained for 14 days. Another cannula, where insulin was later injected, was also implanted at this time. One group of rats receiving saline was pair-fed with the leptin treated group, while the other was allowed to eat ad libitum. On the last day, after a fasting period of 12 h, 10 mU of insulin (Novo Nordisk Pharma, Madrid, Spain) or vehicle was injected *icv* and the rats sacrificed 2 h later. This resulted in the following groups (*n* = 5 per group): chronic vehicle plus acute vehicle (control, C), chronic vehicle plus insulin (insulin, CI), pair-fed with chronic vehicle and acute vehicle (pair-fed, PF), pair-fed with chronic vehicle plus acute insulin (pair-fed plus insulin, PFI), chronic leptin plus acute vehicle (leptin, L) and chronic leptin plus acute insulin (leptin plus insulin, LI). Liver, subcutaneous adipose tissue, epididymal fat and interscapular brown adipose tissue were isolated, and blood collected. Blood was incubated at room temperature for 30 min, centrifuged at 1500 g for 10 min at 4 °C and serum was collected and frozen at –80 °C.

#### 2.2.2. Peripheral Leptin and Central Insulin Administration

Twenty adult male Wistar rats (250 ± 10 g) were anesthetized as stated above and received either saline or leptin (0.2 mg/kg/day) via a subcutaneously implanted Alzet osmotic mini-pump along the back of the rats. After 14 days, 10 mU of insulin (in 7 μL PBS) or PBS alone was injected *icv* and the rats sacrificed by decapitation 2 h later. This resulted in the following groups (*n* = 5 per group): chronic peripheral vehicle plus acute central vehicle (PC, control), chronic peripheral vehicle plus acute central insulin (PCI, insulin), chronic peripheral leptin plus acute central vehicle (PL, leptin) and chronic peripheral leptin plus acute central insulin (PLI, leptin plus insulin).

### 2.3. Serum and Tissue Non-Esterified Fatty Acids (NEFA) and Triglyceride Levels

Concentrations of non-esterified fatty acids (NEFA) and triglycerides were determined by using commercially available kits (Wako Chemicals, Neuss, Germany and Spinreact, St. Esteve de Bas, Spain), following the manufacturer′s instructions. For determination of hepatic concentration of these metabolites, an extraction of total lipids was performed following the method of Folch et al. [28]. The average coefficients of variation for both kits were lower than 10%.

### 2.4. Enzyme-Linked Immunosorbent Assays (ELISAs)

#### 2.4.1. Phosphorylation of Insulin Receptor (IR)

Lysates for enzyme-linked immunosorbent assay (ELISA) methods were obtained using a lysis buffer pH 7.6 with EDTA-free protease inhibitors (Roche Diagnostics, Mannheim, Germany) and Na_3_VO_4_ (Merck, Darmstadt, Germany), as phosphatase inhibitor, and a Retsch ball mills homogenizer (Retsch GmbH, Haan, Germany) at a vibrational frequency of 30 Hz during 1 min. The phosphorylation of this receptor was measured by using an ELISA kit from Assay Solution (Woburn, MA, USA). Hepatic lysates were incubated for 2 h at room temperature (RT) and after washing, a detection antibody coupled to biotin was incubated in the microplate during 2 h at the same temperature. A complex streptavidin-horseradish peroxidase (HRP) was added and finally, after washing, tetramethylbenzidine was incubated until the lecture of the absorbance at 450 nm. The intra- and inter-assay coefficients of variation were lower than 10%.

#### 2.4.2. Phosphorylation of Insulin-Like Growth Factor I Receptor (IGF-IR)

The ELISA (Cell Signaling Technology, Danvers, MA, USA) detects levels of insulin-like growth factor I receptor (IGF-I receptor) β protein when phosphorylated at Tyr1131 residue. Briefly, tissue lysates were incubated for 2 h at 37 °C in a microplate coated with pTyr1131-IGF-IRβ antibody. After washing, a detection antibody was added and incubated at 37 °C for 1 h, the microplate was washed again and then an HRP-linked secondary antibody was added and incubated at the same temperature for 30 min. Finally, the plates were washed again and substrate added until the results were read at 450 nm. The coefficients of variation were lower than 10% and samples were within the linear range of the assay.

#### 2.4.3. Serum Growth Hormone (GH) Levels

Peripheral GH concentrations were measured using an ELISA kit from Merck (Darmstadt, Germany) following the commercial instructions. Coefficients of variations were lower than 10%.

### 2.5. Enzyme Activity Assays

#### 2.5.1. Glucose-6-Phosphate Dehydrogenase

Activity of this dehydrogenase [EC 1.1.1.49] was assayed with a kit of Merck, following the manufacturer′s recommendations. Briefly, after homogenization of 20 mg of liver in phosphate-buffered saline (PBS) and subsequent centrifugation, diluted supernatants were incubated at 37 °C with the master reaction mix and absorbance measured at 450 nm.

#### 2.5.2. Malic Enzyme (ME)

Activity of this enzyme [EC 1.1.1.40] was determined according to the method of Geer et al. [29]. Diluted supernatants were incubated at 25 °C with a triethanolamine buffer, malic acid and NADP and absorbance at 340 nm monitored.

### 2.6. Extraction of RNA and Quantitative Real-Time Polymerase Chain Reaction (qRT-PCR) Analysis

Total RNA was extracted from the hypothalamus, pituitary gland, subcutaneous adipose tissue and liver according to the tri-reagent protocol [30]. The reverse transcription reaction was performed on 2 μg of RNA. Real-time PCR was performed in an ABI Prism 7000 Sequence Detection System (Life Technologies) using a TaqMan PCR Master Mix and thermocycler parameters recommended by the manufacturer. PCRs were performed in a volume of 50 μL, containing 25 μL of the reverse transcription reagents. TaqMan gene expression assays were used for acetyl-CoA carboxylase beta (ACCβ, Rn00588290_m1), carnitine palmitoyl transferase 1a (CPT1a, Rn001475546_m1), FAS (Rn00569117_m1), growth hormone (GH, Rn01495894_g1), SRIF (Rn00561967_m1) and SRIF receptor subtype 2 (sst2, Rn00571116_m1). Relative gene expression comparisons were carried out using an invariant endogenous control (actin, Rn00667869_m1). The ΔΔCT method was used for relative quantification.

### 2.7. Western Blotting

Thirty mg of liver and subcutaneous adipose tissue were homogenized on ice in 500 μL of lysis buffer pH 7.6 containing 50 mmol/L HEPES, 10 mM EDTA, 50 mmol/L sodium pyrophosphate, 100 mmol/L NaF, 10 mmol/L Na_3_VO_4_, 1% Triton X-100, 2 mmol/L phenylmethylsulfonyl fluoride, 10 μg/mL leupeptin and 10 μg/mL aprotinin. Blots were performed using an antibody against GH receptor (GHR, Cell Signaling Technology). The protein bands were detected by chemiluminiscence using an ECL system. Quantification of the bands was carried out by densitometry using a ImageQuant LAS4000 mini-TL Software (GE Healthcare Europe GmbH, Munich, Germany). Gel loading variability for GHR was normalized with actin (Thermo Scientific, Fremont, CA).

### 2.8. Multiplexed Bead Immunoassay

Phosphorylated levels of Akt on threonine 308 (pThr308Akt), extracellular signal-regulated kinase (ERK) on threonine 185 and tyrosine 187 (pThr185Tyr187ERK), phosphorylation of S6 kinase (S6K) on threonine 389 (pThr389S6K) and signal transducer and activator of transcription (STAT)5 on tyrosine 694 (pTyr694STAT5) and total levels of these targets were determined in hepatic homogenates by multiplexed bead immunoassays (Milliplex, Merck and Bio-Plex, Bio-Rad Laboratories, Hercules, CA, USA) as previously described [31]. Levels of Akt phosphorylated on threonine 308 and serine 473, pThr389S6K and total levels of these targets were measured in subcutaneous adipose tissue. A minimum of 50 beads per parameter were analyzed in the Bio-Plex suspension array system 200 (Bio-Rad Laboratories, Madrid, Spain). Raw data (median fluorescence intensity, MFI) were analyzed with the Bio-Plex Manager Software 4.1 (Bio-Rad Laboratories). Mean intra-assay variation was 8.2% and mean inter-assay variation was 12.6%. Phosphorylated targets were normalized with their respective total forms.

### 2.9. Statistical Analysis

Data are presented as mean ± standard error of the mean (SEM). Statistical comparison to assess a possible interaction between leptin and insulin infusion was performed by two-way analysis of variance (ANOVA). Statistical analysis of all data was carried out by one-way ANOVA followed by a Bonferroni’s test. Values were considered significantly different when the *p* value was less than 0.05. Statistical analyses were performed using Statview software (Statview 5.01, SAS Institute, Cary, NC, USA).

## 3. Results

### 3.1. General Characteristics of Experimental Groups

Body weight was recorded to corroborate that leptin infusion affected this parameter. On the eighth day of treatment, body weight was reduced in L and PF groups with respect to controls (Figure 1).

Whole body weight and body weight gain were reduced in both PF and L, with these reductions being greater in L. Liver and soleus content showed no differences among the experimental groups and the gastrocnemius weight presented an increase in L only with respect to the PF group. Percentage of epididymal, inguinal and mesenteric depot masses were decreased in both PF and L, with this reduction being greater in L in the epididymal and inguinal fat pads. Percentage of retroperitoneal fat mass was diminished in the L group and interscapular brown adipose tissue mass was reduced in PF rats (Table 1).

### 3.2. Combined Treatment Increases Serum GH Levels

SRIF mRNA levels were increased in the PF groups and reduced by leptin treatment, with an inhibitory effect of insulin in this group (Table 2). Because sst2 modulates the inhibitory effect of SRIF on GH synthesis [32], we analyzed mRNA levels of this receptor subtype in the pituitary gland. We found that both leptin and pair-feeding reduced sst2 mRNA levels, with insulin also having an inhibitory effect in both controls and leptin treated rats. There was an effect of leptin and insulin, with an interaction between these two factors (F = 4.90, *p* < 0.05), on pituitary GH mRNA levels. Leptin alone had no effect, while pair-feeding decreased GH mRNA levels. A central bolus of insulin decreased the expression of this hormone in C but increased it in the L group. Finally, serum GH levels were decreased in PF rats and insulin only increased serum GH in leptin-treated animals (Table 2).

### 3.3. Hepatic GH Signaling is Increased after Insulin Infusion in Leptin-Treated Rats

Hepatic GHR protein levels were not different between the experimental groups (Figure 2A). Phosphorylation of STAT5 was reduced in the PF group and increased in the leptin plus insulin treated rats (Figure 2B; interaction leptin and insulin: F = 32.71, *p* < 0.001). ERK phosphorylation was also reduced in the PF group and increased after insulin in L rats (Figure 2C, interaction leptin and insulin: F = 6.82, *p* < 0.05). Phosphorylation of insulin receptor (IR) was increased in both groups of leptin-treated rats (Figure 2D, interaction leptin and insulin: F = 6.03, *p* < 0.05). Akt phosphorylation was reduced in the PF group and increased after insulin in PF and L groups (Figure 2E, interaction leptin and insulin: F = 6.41, *p* < 0.05). Phosphorylation of S6K was diminished in the PF group and increased in the PFI and LI groups (Figure 2F, interaction leptin and insulin: F = 8.15, *p* < 0.01). Finally, no changes in the activation of IGF-IR were found (Figure 1G).

### 3.4. Leptin Modifies the Effect of Insulin on Markers of Hepatic Lipid Metabolism

CPT1a mRNA levels were augmented in the PF and L groups and insulin decreased these levels only in L (Figure 3A; interaction leptin and insulin: F = 4.41, *p* < 0.05). Relative mRNA content of ACCβ and FAS were reduced in the PF groups (Figure 3B,C, respectively). There was no effect of insulin on ACCβ in any group (Figure 3B), whereas mRNA levels of FAS increased after an insulin bolus in L rats (Figure 3C; interaction leptin and insulin: F = 8.59, *p* < 0.01). The activity of G6PD was augmented after a bolus of insulin in L rats (Figure 3D; interaction leptin and insulin: F = 17.63, *p* < 0.001). When ME activity was analyzed, it was found to be increased in rats treated with leptin plus insulin (Figure 3E).

### 3.5. Insulin Normalizes the Reduction in Serum Lipid Levels Induced by Leptin

Serum NEFA levels were decreased in L and returned to control values after insulin injection in these rats (Figure 4A). However, there was no effect of insulin injection in C or PF groups. Serum triglyceride concentrations were reduced in the L group and insulin injection normalized these levels (Figure 4B, interaction leptin and insulin: F = 11.08, *p* < 0.001). No effect of insulin in C or PF rats was observed. Moreover, NEFA and triglyceride levels were lower in leptin treated rats compared to the PF group, suggesting that these modifications were not due to the reduction in food intake induced by leptin.

The reduction in serum lipid levels after leptin infusion does not appear to be due to its storage in the liver as hepatic NEFA concentrations were unchanged in L with respect to controls (Figure 4C) and triglyceride concentrations in the liver were decreased by leptin treatment, with no effect of insulin in any group (Figure 4D).

### 3.6. Effect of Leptin and Insulin Infusion in Subcutaneous Fat

The weight of the subcutaneous adipose tissue depot was diminished in pair-fed and leptin-treated rats (Figure 5A). Akt phosphorylation on Thr308 or Ser473 residues (Figure 5B,C, respectively) and phosphorylation of S6K were decreased in the PF and LI groups (Figure 5D) with no effect of insulin in any group. Relative FAS mRNA levels were increased in the central administration of insulin (CI) group (Figure 5E, interaction leptin and insulin: F = 17.49, *p* < 0.001).

### 3.7. Effect of Peripheral Leptin Infusion

Peripheral leptin treatment and/or central insulin injection did not modify serum GH concentrations (Figure 6A). When hepatic GH-related signaling was analyzed, no differences in STAT5 phosphorylation were found (Figure 6B) and activation of ERK was increased after peripheral leptin infusion and it was not modified by central insulin (Figure 6C). Phosphorylation of Akt and S6K in the liver (Figure 6D,E, respectively), as well as serum NEFA levels were unchanged in all experimental groups (Figure 6F).

## 4. Discussion

The present study was conceived to examine the effect of an increase in brain leptin and insulin on hepatic lipid metabolism and the possible relationship with changes in the GH axis. We found that leptin reduced serum lipid levels, possibly through the stimulation of hepatic lipolysis, whereas central insulin infusion normalized the serum lipid profile in leptin-infused rats, at least in part, by increasing hepatic expression of enzymes of lipid anabolism. These changes in serum lipids do not seem to be caused by the contribution of adipose tissue, as insulin-related signaling and lipogenesis are dramatically diminished in this tissue in these rats. We also found that central infusion of leptin plus insulin increased peripheral GH levels and hepatic related signaling, which could explain the subsequent increase in the expression and/or activity of lipogenic enzymes in this experimental group. This increase in serum GH, due to the augment in GH mRNA levels, is associated with a reduction in mRNA levels of hypothalamic SRIF and pituitary sst2.

It is interesting to note that insulin increases GH mRNA levels when leptin is previously administered centrally, as insulin has been reported to improve leptin actions due the crosstalk of these molecules [13] and here it is associated with a remarkable reduction in SRIF mRNA levels. We also observed a decrease in sst2 expression in the pituitary, a SRIF subtype receptor that mediates inhibitory actions of this neuropeptide on GH synthesis [33] in response to insulin in leptin-treated rats that may explain, at least in part, the increase in GH expression found in this study. The increase in serum GH could be due not only to the augment in GH expression, but also due to a stimulatory effect of leptin on secretion combined with reduced SRIF tone, as previously reported [34]. In addition, GH-releasing hormone could contribute to the increase in serum GH in these rats, as central leptin infusion augments hypothalamic GH-releasing hormone in parallel with changes in GH levels [12].

The rise in hepatic CPT1a mRNA levels in leptin-treated rats indicates that this hormone could be involved in the stimulation of lipolysis. Indeed, leptin increases mitochondrial palmitoyl-CoA oxidation rate in liver, which is parallel to the reduction in triglyceride content [35]. Conversely, increased insulin signaling stimulates lipogenesis in liver [36]. Here we report an increase in FAS mRNA levels and Akt activation in response to a central insulin bolus only in rats treated with leptin. In this context, it has been reported that central leptin activates insulin signaling through activation of the PI3K pathway [37] and the improvement in insulin sensitivity raises lipid synthesis [38]. We found a rise in S6K phosphorylation after insulin infusion to leptin-treated rats that may be related to the increase in lipid synthesis. In fact, insulin stimulates hepatic lipogenesis via its ability to trigger sterol regulatory element-binding protein 1c after S6K activation [39]. Although insulin can bind and activate the IGF-I receptor [40], our results seem to indicate that these effects are exclusively due to IR activation, as no changes in phosphorylation of the IGF-I receptor were found.

We cannot discard direct actions of GH, as it inhibits intrahepatic triglyceride lipolysis [41] and increases expression of lipogenic enzymes in the liver [42]. Insulin may play a role in maintaining GH signaling, as physiological levels of insulin are necessary for normal GH sensitivity, regulating hepatic GH receptor levels and intracellular signaling activation. Thus, short-term insulin exposure augments GH-induced activation of the ERK pathway [43] and ERK phosphorylation mediates the expression of FAS [44]. The augmentation in STAT5 activation in the liver of insulin-treated rats is parallel to the serum GH levels and the responses to GH in the liver are mainly mediated by STAT5 phosphorylation [45]. Although synergistic actions of GH and insulin on STAT5 remain unclear, lower activation of STAT5 in a knockout of insulin receptor has been reported [46]. Moreover, transfection of a liver cell line with mir-465 mimics, which may affect genes of PI3K-Akt pathway, reduces STAT5 phosphorylation upon GH stimulation [47].

The provision of NADPH is essential for fatty acid synthesis. The two main enzymes that generate reducing power for this anabolic process are G6PD and malic enzyme. Parallel changes in mRNA levels and activity of hepatic proteins involved in lipogenesis and enzymes that generate reducing power for these anabolic processes have been described [48]. Insulin stimulates transcription of malic enzyme in the liver and situations of insulin resistance or diabetes reduce G6PD activity [49]. Data on GH effects show that its administration increases the activity of G6PD in rat liver [50]. These changes in hepatic lipid metabolism appear to be associated with the serum lipid profile in these animals. In fact, there is a positive association between dyslipidemia and non-alcoholic fatty liver disease [51] and modification of lipogenic enzyme expression in murine models demonstrates an association with serum lipid levels [52]. However, other tissues such as skeletal muscle could be involved in the reduction of serum lipids as we reported an increase in CPT1b expression in the gastrocnemius of leptin-treated rats [53].

The variations in the serum lipid levels of the leptin- and/or insulin-treated rats might be attributed to an increase in hepatic lipid accumulation; however, we found a reduction in hepatic triglyceride levels in the leptin-treated groups. This could probably be related to changes in sensitivity to insulin, as reduced triglyceride stores have been directly associated with increased hepatic PI3K activity [54]. In addition, we cannot discard a direct action of GH, due to its essential role in triglyceride export from the liver [55]; indeed, hepatic GHR deletion has been recently shown to lead to liver steatosis and adenoma formation [56]. Here we demonstrate modifications in the GH axis that could be involved in the observed changes in lipids. Additionally, future research should determine the long-term effects of an increase in the central bioavailability of insulin and changes in serum GH on the generation of steatohepatitis and non-alcoholic fatty liver disease, since GH seems to play an important role in the progression of this disease, together with insulin resistance [57].

We also analyzed insulin-related signaling and FAS expression in subcutaneous adipose tissue, which was reduced in all groups of pair-fed rats, treated or not with insulin. The reduction in insulin signaling and FAS mRNA levels in these experimental groups may indicate that this tissue does not contribute substantially to the modifications in serum lipid profile described here. We have previously reported that serum leptin levels were similar after central and peripheral leptin infusion [27]. We cannot distinguish between central and circulating leptin effects, but here we show that central leptin infusion is more potent than peripheral administration in modulating hepatic insulin signaling after insulin bolus. In this way, central leptin administration produces changes in serum levels of some hormones associated with increased insulin sensitivity, whereas peripheral administration did not produce the same variations [58]; moreover, the effects on the reduction in food intake and body weight also differed [59].

A caveat should be taken into consideration when evaluating these results. We have not assessed total body composition, which could influence lipid metabolism and the lipid profile. Evidence indicates that central leptin infusion modifies fat depots [60]. Body composition measurements provide an accurate determination of fat and lean percentages of the animal in these studies. We have included the percentage of weight measurements in fat and muscle in different locations. Other measures not determined here, such as gut weight reduction after leptin infusion [61], may contribute to the least weight gain in leptin-treated animals.

## 5. Conclusions

Our data indicate that the reduction in serum lipid levels after chronic leptin infusion is associated with changes in hepatic lipid metabolism. As summarized in Figure 7, our findings support the notion that central inhibition of SRIF action in leptin plus insulin-treated rats may be involved in the increase in GH levels and could increase lipid anabolism by potentiating hepatic insulin and GH sensitivity. These results suggest that changes in brain leptin and insulin signaling could modify the peripheral lipid profile through modulation of hepatic metabolism.

## Figures and Tables

**Figure 1 cells-10-00581-f001:**
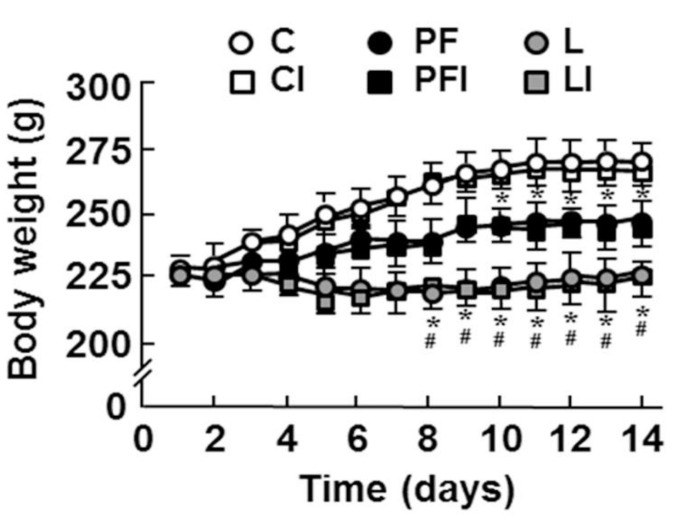
Mean body weight measurements throughout the study in control rats (C); rats receiving acute central administration of insulin (CI); pair-fed rats (PF); pair-fed rats treated with insulin (PFI); rats treated with chronic *icv* leptin infusion (L) and rats treated with leptin plus insulin (LI). * *p* < 0.05 vs. C groups, # *p* < 0.05 vs. PF groups.

**Figure 2 cells-10-00581-f002:**
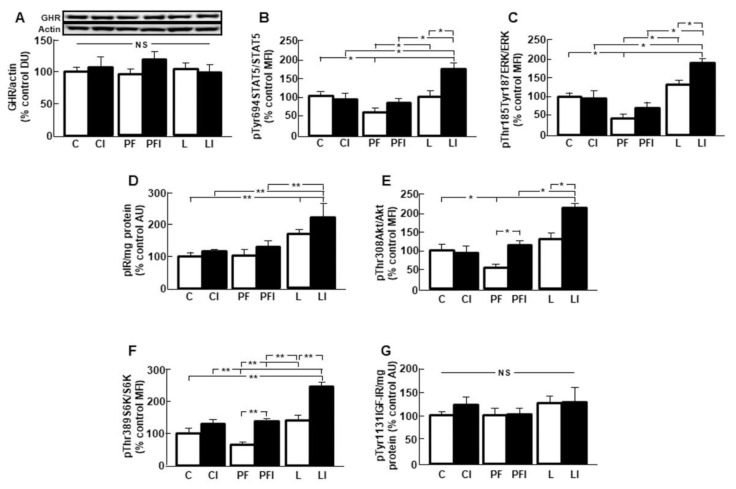
Effects of leptin and insulin infusion on growth hormone (GH)-related signaling in the liver. (**A**) Relative GH receptor (GHR) protein levels in control rats (C); rats receiving acute central administration of insulin (CI); pair-fed rats (PF); pair-fed rats treated with insulin (PFI); rats treated with chronic *icv* leptin infusion (L) and rats treated with leptin plus insulin (LI). (**B**) Relative phosphorylated (p) signal transducer and activator of transcription (STAT)5 on tyrosine 694 (pTyr694STAT5) protein levels (**C**) Relative phosphorylated extracellular signal-regulated kinases (ERK) on threonine 185/tyrosine 187 (pThr185/Tyr187ERK) protein levels. (**D**) Relative protein levels of phosphorylated insulin receptor (pIR). (**E**) Relative phosphorylated Akt on threonine 308 (pThr308Akt) protein levels. (**F**) Relative phosphorylated S6 kinase on threonine 389 (pThr389S6K) protein levels. (**G**) Relative phosphorylated insulin-like growth factor I receptor on Tyr 1131 (pTyr1131IGF-IR). Data are presented as means ± SEM. *n* = 5. AU, absorbance units; DU, densitometry units; MFI, median fluorescent intensity, NS, non-significant. * *p* < 0.05, ** *p* < 0.01.

**Figure 3 cells-10-00581-f003:**
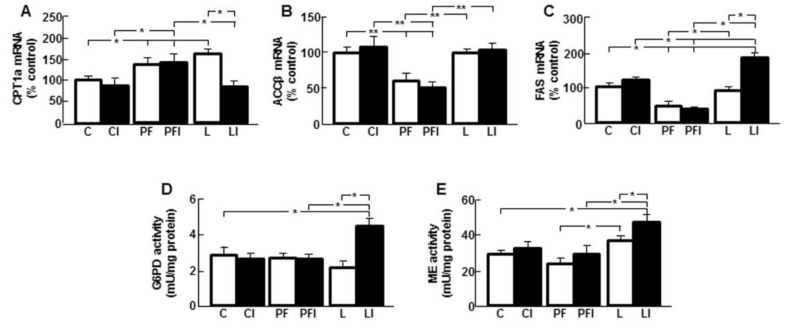
Central leptin and insulin infusion change the mRNA levels and activity of enzymes involved in hepatic lipid metabolism. (**A**) Relative carnitine palmitoyl transferase (CPT)1a mRNA content in the liver of control rats (C); rats receiving acute central administration of insulin (CI); pair-fed rats (PF); pair-fed rats treated with insulin (PFI); rats treated with chronic *icv* leptin infusion (L) and rats treated with leptin plus insulin (LI). (**B**) Relative acetyl-CoA carboxylase (ACC)β mRNA content. (**C**) Relative fatty acid synthase (FAS) mRNA content. (**D**) Glucose-6-phosphate dehydrogenase (G6PD) activity. (**E**) Malic enzyme (ME) activity. Data are presented as means ± SEM. *n* = 5. * *p* < 0.05, ** *p* < 0.01.

**Figure 4 cells-10-00581-f004:**
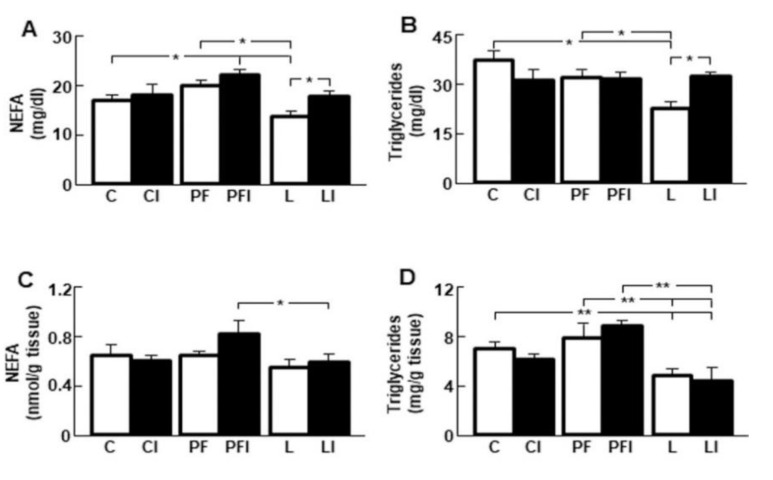
Insulin infusion returns to control values the serum lipid profile in leptin-treated rats back to those seen in control animals. (**A**) Serum non-esterified fatty acid (NEFA) levels in control rats (C); rats receiving acute central administration of insulin (CI); pair-fed rats (PF); pair-fed rats treated with insulin (PFI); rats treated with chronic *icv* leptin infusion (L) and rats treated with leptin plus insulin (LI). (**B**) Serum triglyceride levels. (**C**) Hepatic NEFA concentrations. (**D**) Hepatic triglyceride concentrations. Data are presented as means ± SEM. *n* = 5. * *p* < 0.05, ** *p* < 0.01.

**Figure 5 cells-10-00581-f005:**
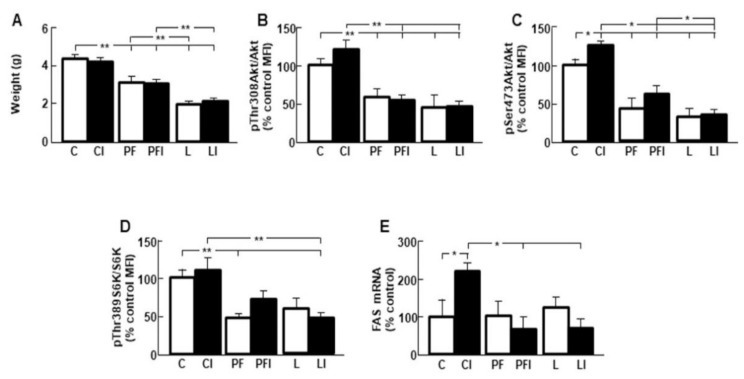
Changes in subcutaneous adipose tissue. (**A**) Weight in control rats (C); rats receiving acute central administration of insulin (CI); pair-fed rats (PF); pair-fed rats treated with insulin (PFI); rats treated with chronic *icv* leptin infusion (L) and rats treated with leptin plus insulin (LI). (**B**) Relative phosphorylated (p) Akt on theonine 308 (pThr308Akt) protein levels. (**C**) Relative phosphorylated Akt on serine 473 (pSer473Akt) protein levels. (**D**) Relative phosphorylated S6 kinase on threonine 389 (pThr389S6K) protein levels. (**E**) Relative fatty acid synthase (FAS) mRNA levels. Values are means ± SEM. *n* = 5. MFI, median fluorescent intensity. * *p* < 0.05, ** *p* < 0.01.

**Figure 6 cells-10-00581-f006:**
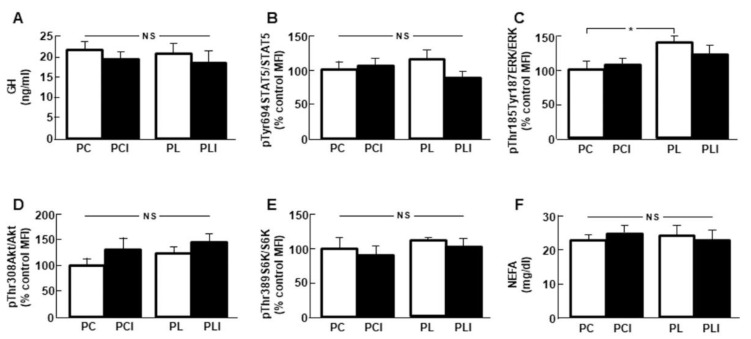
Effects of peripheral leptin administration. (**A**) Serum growth hormone (GH) levels in control rats (PC); rats receiving acute central administration of insulin (PCI); rats treated with chronic peripheral leptin administration (PL) and rats treated with chronic peripheral leptin plus central insulin (PLI). (**B**) Relative phosphorylated (p) signal transducer and activator of transcription (STAT)5 on tyrosine 694 (pTyr694STAT5) protein levels in the liver. (**C**) Relative phosphorylated extracellular signal–regulated kinases (ERK) on threonine 185/tyrosine 187 (pThr185/Tyr187ERK) protein levels in the liver. (**D**) Relative phosphorylated Akt on threonine 308 (pThr308Akt) protein levels in the liver. (**E**) Relative phosphorylated S6 kinase on threonine 389 (pThr389S6K) protein levels. (**F**) Serum non-esterified fatty acid (NEFA) levels. Data are presented as means ± SEM. *n* = 5. MFI, median fluorescent intensity, NS, non-significant. * *p* < 0.05.

**Figure 7 cells-10-00581-f007:**
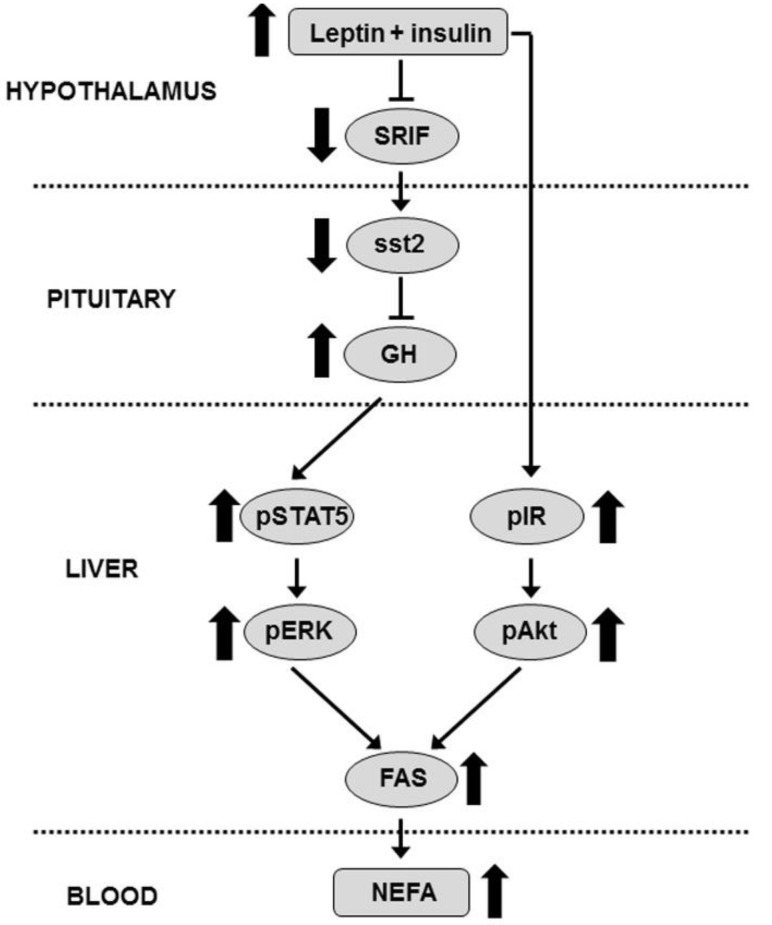
Proposed mechanism of action. Combined treatment with leptin and insulin diminishes SRIF and sst2 expression, increasing GH mRNA levels. The increase in serum GH levels augments STAT5 and ERK phosphorylation, which together with the increase in activation of insulin signaling induce FAS expression, augmenting serum NEFA levels in these rats. Akt, protein kinase B; ERK, extracellular signal-regulated kinase; FAS, fatty acid synthase; GH, growth hormone; IR, insulin receptor; NEFA, non-esterified fatty acids; p, phosphorylated; SRIF, somatostatin; sst2, SRIF subtype receptor 2; STAT5, signal transducer and activator of transcription factor 5.

**Table 1 cells-10-00581-t001:** General characteristics of experimental groups.

Parameter	C	CI	PF	PFI	L	LI
Body weight (g)	267 ± 8	261 ± 7	253 ± 6 *	250 ± 5 *	228 ± 3 *^#^	227 ± 5 *^#^
Body weight gain (g)	43.4 ± 3.2	43.0 ± 3.6	25.1 ± 2.2 **	25.3 ± 2.0 **	2.8 ± 1.5 **^##^	2.9 ± 1.4 **^##^
Gastrocnemius (%)	0.87 ± 0.05	0.90 ± 0.06	0.75 ± 0.03	0.75 ± 0.04	1.12 ± 0.07 ^#^	1.14 ± 0.08 ^#^
Soleus (%)	0.07 ± 0.01	0.06 ± 0.01	0.07 ± 0.01	0.07 ± 0.01	0.08 ± 0.02	0.09 ± 0.02
Liver weight (%)	3.53 ± 0.19	3.54 ± 0.18	3.58 ± 0.24	3.59 ± 0.22	3.80 ± 0.32	3.83 ± 0.26
Epididymal fat (%)	1.39 ± 0.06	1.43 ± 0.10	1.03 ± 0.09 **	1.07 ± 0.08 **	0.65 ± 0.10 **^##^	0.66 ± 0.13 **^##^
Inguinal fat (%)	1.06 ± 0.03	1.08 ± 0.05	0.88 ± 0.05 *	0.87 ± 0.04 *	0.66 ± 0.06 *^#^	0.65 ± 0.07 *^#^
Mesenteric fat (%)	1.12 ± 0.14	1.15 ± 0.13	0.71 ± 0.06 *	0.74 ± 0.08 *	0.67 ± 0.08 *	0.65 ± 0.10 *
Retroperitoneal fat (%)	0.37 ± 0.04	0.37 ± 0.05	0.35 ± 0.03	0.34 ± 0.04	0.28 ± 0.02 *	0.29 ± 0.02 *
iBAT (%)	0.26 ± 0.03	0.26 ± 0.02	0.17 ± 0.02 *	0.18 ± 0.03 *	0.24 ± 0.03	0.25 ± 0.04

Value are means ± SEM of five animals. C, control rats; CI, rats receiving acute central administration of insulin; PF, pair-fed rats; PFI, pair-fed rats treated with insulin; L, rats treated with chronic icv leptin infusion and LI, rats treated with leptin plus insulin. iBAT, interscapular brown adipose tissue. * *p* < 0.05, ** *p* < 0.01 vs. C; ^#^
*p* < 0.05, ^##^
*p* < 0.01 vs. PF.

**Table 2 cells-10-00581-t002:** Effects of central leptin plus insulin on components of the growth hormone (GH) axis.

Parameter	C	CI	PF	PFI	L	LI
SRIF mRNA (% control)	100.0 ± 7.7	110.5 ± 8.4	132.5 ± 9.6 *	169.2 ± 8.5 *	46.0 ± 3.3 *	21.5 ± 1.7 *^#^
sst2 mRNA (% control)	100.0 ± 9.0	19.7 ± 5.6 *	45.9 ± 11.0 *	42.9 ± 12.8 *	42.5 ± 6.4 *	20.3 ± 3.6 *^#^
GH mRNA (% control)	100.0 ± 10.6	64.0 ± 7.8 *	60.0 ± 6.6 *	80.5 ± 21.1	85.9 ± 16.9	141.6 ± 11.3 *^#^
GH (ng/mL)	16.8 ± 2.4	22.4 ± 2.5	13.7 ± 2.0 *	16.2 ± 3.0	32.9 ± 4.6	45.9 ± 5.8 *^#^

Value are means ± SEM of five animals. C, control rats; CI, rats receiving acute central administration of insulin; PF, pair-fed rats; PFI, pair-fed rats treated with insulin; L, rats treated with chronic icv leptin infusion and LI, rats treated with leptin plus insulin. SRIF, somatostatin; sst2, somatostatin receptor subtype 2. * *p* < 0.05 vs. C, ^#^
*p* < 0.05 vs. L.

## Data Availability

All relevant data are included within the manuscript.

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
