# Peer review of "Cerebral Insulin Bolus Revokes the Changes in Hepatic Lipid Metabolism Induced by Chronic Central Leptin Infusion"

_cells, 2021, doi:10.3390/cells10030581_

Round 1
Reviewer 1 Report
In this paper, by using a chronicle leptin infusing model plus the infusion of insulin in rats, the authors studied the actions of leptin and insulin on hepatic lipid metabolism. The authors are trying to conclude that an increase in insulin in leptin-treated rats can act on the hepatic lipid metabolism through 34 activation of the GH axis.
There are a few concerns need to be addressed for this paper:
1) The authors did not clearly present the data about changes in GH in pituitary nor the level of pituitary somatostatin receptor-2 mRNA levels. There were data mentioned in lines 213-219, however that is not sufficient, specially if the authors want to make final conclusion to indicate the involvement of GH in this action of insulin. This set of data (described in line 213-219) need to be clearly presented with data of GH mRNA before and after insulin treatment and also the sst2 mRNA levels before and after insulin treatment.
2) The animals were treated with leptin for 14 days, were there any changes in the body weight ? or body composition? The weight of subcutaneous adipose tissue were presented in Figure 4, however this is not sufficient. The whole body weight, liver weight, visceral adipose tissue and also brown adipose tissue weight need to be monitored and reported. It would also be good to perform a body composition and report the % body fat.
3) Some minor details: for sections 2.4.1 and 2.4.2, please provide details for how the lysate were prepared? homogenized? with protease and phosphatase inhibitors?
5) The paper would benefit from check from a professional editor for gramma and language.
ie:
--for line 299:
3.4. Insulin normalizes the reduction in lipid serum levels induced by leptin
Would be better: 3.4. Insulin normalizes the reduction in serum lipid levels induced by leptin
--for line 335:
"Figure 3. Insulin infusion returns to control values the serum lipid profile in leptin-treated rats".
Would be better: Figure 3. Insulin infusion returns the serum lipid profile in leptin-treated rats back to those seen in control animals.
5) If possible, the authors should consider adding a diagram to illustrate the mechanism of how this action of insulin works through the GH axis.
Author Response
Reviewer 1
In this paper, by using a chronicle leptin infusing model plus the infusion of insulin in rats, the authors studied the actions of leptin and insulin on hepatic lipid metabolism. The authors are trying to conclude that an increase in insulin in leptin-treated rats can act on the hepatic lipid metabolism through 34 activation of the GH axis.
There are a few concerns need to be addressed for this paper:
1) The authors did not clearly present the data about changes in GH in pituitary nor the level of pituitary somatostatin receptor-2 mRNA levels. There were data mentioned in lines 213-219, however that is not sufficient, specially if the authors want to make final conclusion to indicate the involvement of GH in this action of insulin. This set of data (described in line 213-219) need to be clearly presented with data of GH mRNA before and after insulin treatment and also the sst2 mRNA levels before and after insulin treatment.
Response: The authors thank the Reviewer for his/her constructive comments. These results have been included in Table 2, 3.2. subsection, page 5.
2) The animals were treated with leptin for 14 days, were there any changes in the body weight ? or body composition? The weight of subcutaneous adipose tissue were presented in Figure 4, however this is not sufficient. The whole body weight, liver weight, visceral adipose tissue and also brown adipose tissue weight need to be monitored and reported. It would also be good to perform a body composition and report the % body fat.
Response: Body weight, liver weight, visceral adipose tissue and brown adipose tissue weight were monitored and the results are now presented in Table 1, 3.1. subsection, page 5. We certainly agree with the Reviewer that the evaluation of body composition would be very interesting. Unfortunately, it would imply carrying out a new set of experiments.
3) Some minor details: for sections 2.4.1 and 2.4.2, please provide details for how the lysate were prepared? homogenized? with protease and phosphatase inhibitors?
Response: We have provided more detail for the obtention of lysates. This information is now included in the first paragraph of subsection 2.4.1., page 3.
Lysates for ELISA were obtained using a lysis buffer pH 7.6 with EDTA-free protease inhibitors (Roche Diagnostics, Mannheim, Germany) and Na3VO4 (Merck, Darm-stadt, Germany) as a phosphatase inhibitor, and a Retsch ball mill homogenizer (Retsch GmbH, Haan, Germany) at a vibrational frequency of 30 Hz during 1 min.
5) The paper would benefit from check from a professional editor for gramma and language.
ie:
--for line 299:
3.4. Insulin normalizes the reduction in lipid serum levels induced by leptin
Would be better: 3.4. Insulin normalizes the reduction in serum lipid levels induced by leptin
Response: The title of this subsection has been modified.
--for line 335:
"Figure 3. Insulin infusion returns to control values the serum lipid profile in leptin-treated rats".
Would be better: Figure 3. Insulin infusion returns the serum lipid profile in leptin-treated rats back to those seen in control animals.
Response: This sentence has been corrected.
5) If possible, the authors should consider adding a diagram to illustrate the mechanism of how this action of insulin works through the GH axis.
Response: We have included a diagram with the proposed mechanism of action (figure 6, page 11).

Reviewer 2 Report
GENERAL COMMENT
As a clinician I am much interested in the translational value of this study and would like to ask these authors to speculate on the possible implication(s) of their findings as applied to the human nonalcoholic fatty liver (NAFLD) arena (Nat Rev Gastroenterol Hepatol. 2009 Apr;6(4):236-47. Neuroendocrinology. 2017;104(4):364-381. Int J Mol Sci. 2019 Mar 15;20(6):1317. Int J Mol Sci. 2019 Jun 11;20(11):2841). This may be articulated both along the pathway of insulin resistance as well as regarding the role of GH in the development and progression of NAFLD
SPECIFIC COMMENT
In humans, the ultimate mechanisms which are eventually conducive to the development of visceral adiposity and insulin resistance are far from being clear and indeed it is possible that central dysregulation does play a major role (J Clin Med. 2021 Jan 31;10(3):492). Confirming this, a recent study found that that high brain insulin sensitivity anticipates weight loss during lifestyle intervention and is associated with a favorable body fat distribution; additionally high brain insulin sensitivity is also associated with less regain of fat mass during a nine-year follow-up (Kullmann S, Valenta V, Wagner R, Tschritter O, Machann J, Häring HU, Preissl H, Fritsche A, Heni M. Brain insulin sensitivity is linked to adiposity and body fat distribution. Nat Commun. 2020;11:1841).
Along the same line, Growth Hormone (GH) deficiency is deemed to be a risk factor for the development of NASH, the more rapidly evolutive NAFLD variant (Roman Liebe, Irene Esposito, Hans H. Bock, Stephan vom Dahl, Jan Stindt, Ulrich Baumann, Tom Luedde, Verena Keitel,Diagnosis and management of secondary causes of steatohepatitis, Journal of Hepatology, 2021,ISSN 0168-8278, https://doi.org/10.1016/j.jhep.2021.01.045. (JCI Insight. 2020 Aug 20;5(16):e140134. Hepatol Int. 2018 Sep;12(5):474-481. Horm Metab Res. 2018 Mar;50(3):250-256. Intern Med. 2017;56(5):473-480. Pediatr Transplant. 2016 Dec;20(8):1157-1163. . Growth Horm IGF Res. 2014 Oct;24(5):174-9. Hepatology. 2014 May;59(5):1668-70. PLoS One. 2012;7(8):e44136).
Author Response
Reviewer 2
GENERAL COMMENT
As a clinician I am much interested in the translational value of this study and would like to ask these authors to speculate on the possible implication(s) of their findings as applied to the human nonalcoholic fatty liver (NAFLD) arena (Nat Rev Gastroenterol Hepatol. 2009 Apr;6(4):236-47. Neuroendocrinology. 2017;104(4):364-381. Int J Mol Sci. 2019 Mar 15;20(6):1317. Int J Mol Sci. 2019 Jun 11;20(11):2841). This may be articulated both along the pathway of insulin resistance as well as regarding the role of GH in the development and progression of NAFLD
Response: We would like to thank the Reviewer for his/her constructive observations. We have included this interesting point in the next to last paragraph of Discussion (page 10).
“The variations in the serum lipid levels of the leptin- and/or insulin-treated rats could be attributed to an increase in hepatic lipid accumulation; however, we found a reduction in hepatic triglyceride levels in the leptin-treated groups. This could possibly be related to an increase in insulin sensitivity, as reduced triglyceride stores have been directly associated with in-creased hepatic PI3K activity [54]. In addition, we cannot discard an effect of GH, due to its essential role in triglyceride export from the liver [55]; indeed, hepatic GHR deletion has been recently shown to lead to liver steatosis and adenoma formation [56]. Here we demonstrate modifications in the GH axis that could be involved in the observed changes in lipids. Future research should determine the long-term effects of modifications in the central bioavailability of, or sensitivity to, insulin and changes in serum GH on the generation of steato-hepatitis and nonalcoholic fatty liver disease, since GH together with insulin resistance, appear to play an important role in the progression of this disease [57]”.
SPECIFIC COMMENT
In humans, the ultimate mechanisms which are eventually conducive to the development of visceral adiposity and insulin resistance are far from being clear and indeed it is possible that central dysregulation does play a major role (J Clin Med. 2021 Jan 31;10(3):492). Confirming this, a recent study found that that high brain insulin sensitivity anticipates weight loss during lifestyle intervention and is associated with a favorable body fat distribution; additionally high brain insulin sensitivity is also associated with less regain of fat mass during a nine-year follow-up (Kullmann S, Valenta V, Wagner R, Tschritter O, Machann J, Häring HU, Preissl H, Fritsche A, Heni M. Brain insulin sensitivity is linked to adiposity and body fat distribution. Nat Commun. 2020;11:1841).
Response: Thank you for this comment and these important observations have been included in the third paragraph of the Introduction section (page 2).
“High brain insulin sensitivity is associated with weight loss and related to a favorable fat distribution [9], whereas insulin resistance is linked to visceral adiposity and pathological accumulation of intrahepatic fat content [10]. Our findings indicate that an increase in central insulin bioavailability may modify synthesis and accumulation of fat in the liver, as well as to provoke changes in adipose tissue that could affect systemic lipid concentrations”.
Along the same line, Growth Hormone (GH) deficiency is deemed to be a risk factor for the development of NASH, the more rapidly evolutive NAFLD variant (Roman Liebe, Irene Esposito, Hans H. Bock, Stephan vom Dahl, Jan Stindt, Ulrich Baumann, Tom Luedde, Verena Keitel,Diagnosis and management of secondary causes of steatohepatitis, Journal of Hepatology, 2021,ISSN 0168-8278, https://doi.org/10.1016/j.jhep.2021.01.045. (JCI Insight. 2020 Aug 20;5(16):e140134. Hepatol Int. 2018 Sep;12(5):474-481. Horm Metab Res. 2018 Mar;50(3):250-256. Intern Med. 2017;56(5):473-480. Pediatr Transplant. 2016 Dec;20(8):1157-1163. . Growth Horm IGF Res. 2014 Oct;24(5):174-9. Hepatology. 2014 May;59(5):1668-70. PLoS One. 2012;7(8):e44136).
Response: We have included information pertaining to this point in the fifth paragraph of Introduction.
“GH deficiency and insulin resistance are risk factors for the development of nonalcoholic steatohepatitis and previous reports suggest a relationship between these hormones in this pathology [19,20]. GH regulates numerous metabolic processes in the liver, including lipid metabolism via interactions with leptin and insulin [21,22] and promotion of triglyceride uptake [23]. In fact, maintenance of normal insulinemia seems to be required to maintain physiological hepatic GH sensitivity [24]. GH is also implicated in the generation of NADPH for lipogenesis, as up-regulation of the GH axis correlates with augmented hepatic G6PD expression [25]. Short-term effects of GH are insulin-like, inhibiting lipolysis and stimulating lipogenesis, whereas prolonged elevated concentrations of this hormone provoke insulin resistance related to reduction in phosphatidylino-sitol-3 kinase (PI3K) activation [26]”.

Round 2
Reviewer 1 Report
Thanks to the authors' effort in addressing the reviewer's concern, however the body weight, liver weight and adipose weight do not add up. The authors reported ~43 g, 25 g and 3g changes in body weight during the 15days leptin treatment for the Control, pair fed and leptin treatment group, respectively. However the difference between the weight of liver+adipose tissues do not add up between the groups, what are the other changes that is happening in these animals with different treatment?
1) A complete growth curve for (starting body weight and changes of body weight daily or every other day) this 15 days for the different treatment groups would be necessary.
2) Body composition would be necessary to show the effect of treatment on lean body mass.
These data are essential to show that how the animal models was set up.
Author Response
Reviewer 1
Thanks to the authors' effort in addressing the reviewer's concern, however the body weight, liver weight and adipose weight do not add up. The authors reported ~43 g, 25 g and 3g changes in body weight during the 15 days leptin treatment for the Control, pair fed and leptin treatment group, respectively. However the difference between the weight of liver+adipose tissues do not add up between the groups, what are the other changes that is happening in these animals with different treatment?
Response: We would like to thank the Reviewer for his/her constructive observations.
We have added the weight percentage of different visceral tissues in Table 1 that contribute to final body weight. The decrease in the inguinal, mesenteric and retroperitoneal adipose tissues in pair-fed and leptin-treated may contribute to the differences, although other measures of course would add important information. However, these data are not available in this study. We would also like to point out that as we do not have the change in weights of the different organs/tissues, only the final weight, it is impossible to determine which tissues contribute to the overall change in weight. This limitation has been included in the last paragraph of the Discussion section.
1) A complete growth curve for (starting body weight and changes of body weight daily or every other day) this 15 days for the different treatment groups would be necessary.
Response: We have included a complete growth curve during the 15 days for all experimental groups (Figure 1, page 5).
2) Body composition would be necessary to show the effect of treatment on lean body mass.
Response: We have included the weight percentage of the soleus and gastrocnemius as a measure of lean body mass. We do not have bioimpedance, DEXA or NMR equipment to measure body composition and on the other hand, this would require repeating the study again with new animals. Previous results published in the literature indicate that chronic leptin infusion changes fat mass, but lean mass is scarcely affected or unaffected. In this way, the requested measures in muscle and visceral fat pads are according to these data (Ravussin et al., Mol Metab 2014; 3: 432-440). Nevertheless, we agree with the Reviewer that this information would be interesting and this limitation of the study has been included in last paragraph of the Discussion section (page 11).
These data are essential to show that how the animal models was set up.
